# The podocin V260E mutation predicts steroid resistant nephrotic syndrome in black South African children with focal segmental glomerulosclerosis

Melanie A. Govender[1], June Fabian[2,3], Errol Gottlich[2,4], Cecil Levy[5], Glenda Moonsamy[6], Heather Maher[2], Cheryl A. Winkler [7] & Michèle Ramsay [1]*

In black African children with focal segmental glomerulosclerosis (FSGS) there are high rates of steroid resistance. The aim was to determine genetic associations with apolipoprotein L1 (*APOL1*) renal risk variants and podocin (*NPHS2*) variants in 30 unrelated black South African children with FSGS. Three *APOL1* variants were genotyped and the exons of the *NPHS2* gene sequenced in the cases and controls. *APOL1* risk alleles show a modest association with steroid sensitive nephrotic syndrome (SSNS) and steroid resistant nephrotic syndrome (SRNS). The *NPHS2* V260E variant was present in SRNS cases (V/V = 5; V/E = 4; E/E = 11), and was absent in SSNS cases. Haplotype analysis suggests a single mutation origin for V260E and it was associated with a decline in kidney function over a 60-month period (p = 0.026). The V260E variant is a good predictor of autosomal recessive SRNS in black South African children and could provide useful information in a clinical setting.

[1] Sydney Brenner Institute for Molecular Bioscience and Division of Human Genetics, National Health Laboratory Service and School of Pathology, Faculty of Health Sciences, University of the Witwatersrand, Johannesburg, South Africa. [2] Wits Donald Gordon Medical Centre, University of the Witwatersrand, Johannesburg, South Africa. [3] Division of Nephrology, Department of Internal Medicine, School of Clinical Medicine, Faculty of Health Sciences, University of the Witwatersrand, Johannesburg, South Africa. [4] Department of Paediatrics, University of Pretoria, Pretoria, South Africa. [5] Nelson Mandela Children's Hospital, Division of Nephrology, Department of Paediatrics, School of Clinical Medicine, Faculty of Health Sciences, University of the Witwatersrand, Johannesburg, South Africa. [6] Charlotte Maxeke Johannesburg Academic Hospital, Division of Nephrology, Department of Paediatrics, Faculty of Health Sciences, University of the Witwatersrand, Johannesburg, South Africa. [7] Basic Research Laboratory, Center for Cancer Research, National Cancer Institute, Leidos Biomedical Research, Inc., Frederick National Laboratory, Frederick, MD, USA. *email: michele.ramsay@wits.ac.za

Nephrotic syndrome (NS) is characterised by four clinical features: nephrotic range proteinuria, hypoalbuminemia, oedema and hyperlipidaemia[1]. In children, NS is further categorised by response or lack of response to steroid treatment into 'steroid sensitive nephrotic syndrome' (SSNS) versus 'steroid resistant nephrotic syndrome' (SRNS)[2,3]. In children with NS, focal segmental glomerulosclerosis (FSGS) is the glomerular pathology most commonly associated with progression to end stage kidney disease (ESKD)[4]. A higher incidence of FSGS is reported among children with African ancestry and a more rapid progression of FSGS to ESKD is observed, compared to those with European ancestry[5–7]. The higher prevalence of biopsy-proven FSGS in children and adults, among certain ethnic groups, suggests the role of genetic susceptibility to disease.

Over 50 genes have been associated with familial and monogenic forms of SRNS, with autosomal recessive and dominant modes of inheritance, manifesting in both children and adults. Genes commonly mutated in childhood SRNS are NPHS1 (nephrin), NPHS2 (podocin) and WT1 (Wilms tumour 1), with variants in the NPHS2 gene being the most common cause of SRNS in children[8–11]. A large study by Sadowski et al. (2015), performed on 1783 families (of different ethnic origins, including Africans), identified monogenic autosomal recessive causes of SRNS in 418 of the families. When the age of onset was before 25 years, the following genes were most often implicated: NPHS2, NPHS1 and WT1[8]. Several clinical genetics groups have introduced a next generation sequencing (NGS) approach for panels of genes involved in SRNS and have observed significant mutation profile differences across countries and regions[12–14].

Three coding variants in the apolipoprotein L1 (APOL1) gene are significantly associated with susceptibility to FSGS and a spectrum of chronic kidney disease (CKD) in African Americans[15,16]. These variants include two non-synonymous variants collectively referred to as the G1 risk allele (rs73885319 and rs60910145) and a 6 bp in-frame deletion (rs71785313) referred to as the G2 risk allele[16,17]. While there are few studies exploring these associations in Africa, a strong association between adults carrying two APOL1 risk variants and HIV- associated nephropathy (HIVAN) was demonstrated in South Africa, with an OR of 89 (95% CI, 17.7–912)[18]. However, the APOL1 risk alleles were not significantly associated with FSGS (OR 6.03; 95% CI, 0.08–527), possibly due to the small sample[18]. In African American children with FSGS, 78% were found to carry two APOL1 risk variants with a median age of onset of 12 years[19]. The role of APOL1 risk variants with FSGS in black South African children remains uncertain.

In South Africa, there are limited genetic studies on kidney disease, including FSGS, among children. Asharam et al. (2018)[20] reported on 33 Indian and 31 black children from Kwa-Zulu Natal with SSNS and SRNS. In their study, APOL1 variants were not associated with NS, but the NPHS2 V260E allele was specifically associated with SRNS (with FSGS) in black children; 8 were homozygous for the variant (OR 21; 95% CI, 2.8–960) and no heterozygotes were observed. The NPHS2 V260E variant alters the podocin protein and prevents the movement of the protein to the plasma membrane, affecting the structural integrity of the slit diaphragm[21]. This variant was absent in the Indian children (cases and controls) and absent in all black children with SSNS. One black control was heterozygous for this variant[20].

In South Africa, there are limited resources for treating children who progress to ESKD, rendering this condition fatal for those who are not treated[22]. At least half of the individuals who require ESKD treatment cannot be accommodated, and only 1.5% are predicted to receive renal replacement therapy[23]. Genetic counselling services are available, but limited, and our goal is to develop an affordable assay with a high predictive value to detect children with kidney disease who are unlikely to respond to steroid treatment.

In this study, the aim was to determine genetic associations with APOL1 risk variants and NPHS2 variants in 32 black South African children with biopsy-proven FSGS who receive treatment in Johannesburg, Gauteng Province, South Africa. FSGS is the most common lesion associated with progression to ESKD in children. The majority of black South African children with FSGS do not respond to steroid treatment (SRNS). An assay predicting which children are likely to respond to steroid treatment would be of significant clinical value. Our study identified a common African NPHS2 V260E variant associated with steroid resistant FSGS in black South African children. This variant appears to be associated with a decline in kidney function and was not found in children with steroid sensitive FSGS, indicating that it could be useful in making treatment decisions in a clinical setting. Our study contributes additional information to that of Asharam et al. (2018)[20], preformed on affected children in the Zulu population, by extending the ethnolinguistic origins of the patients and by identifying affected individuals heterozygous for the V260E variant.

## Results

**Study participants.** The files of 48 patients with FSGS were identified from CMJAH and MM and 32 met our study selection criteria. Sixteen participants were excluded due to incomplete data or loss to follow-up. The 32 children with biopsy-proven FSGS had a mean age of onset of 6 years and the numbers that progressed to ESKD (dialysis and transplantation) are shown in Table 1. There were two familial cases of FSGS with affected siblings that included monozygotic twins in one family and affected sisters in the second family. SRNS was more frequent (68.8%) compared to SSNS (31.2%). The patients were of different ethnic origins including: Tsonga, Tswana, Swati, Sotho, Pedi and Zulu.

**APOL1 associations with FSGS.** Among the unrelated FSGS cases, allele frequencies for the APOL1 G1 and G2 risk alleles were 10% and 22%, respectively, with 8% and 13% in the controls (Supplementary Table 1). Table 2 shows that in the SS-FSGS group, 70% carried 1 APOL1 risk variant compared to 26% of the control group (OR 4.63; 95% CI, 1.15–18.5). Among all FSGS patients who did not have the NPHS2 V260E variant, having 1 or 2 APOL1 risk alleles was associated with FSGS (OR 2.97; 95% CI, 1.01–8.75)) and only one child had 2 APOL1 risk alleles.

**Variants in the NPHS2 gene.** NPHS2 exon sequencing was performed in all 32 cases (including sib-pairs). Two missense (A242V and V260E) and five synonymous variants (G34G, S48S, S96S, A318A and L346L) were observed (Table 3). All

### Table 1 Characteristics of black South African children with focal segmental glomerulosclerosis (FSGS)

| Characteristic | SSNS (n = 10) | SRNS (n = 22)[a] | P value** |
|---|---|---|---|
| Sex | | | |
| Male | 4 (40%) | 9 (40.9%) | 0.9999 |
| Female | 6 (60%) | 13 (59.1%) | |
| Mean age of onset | 4.7 | 7.1 | 0.023 |
| ESKD[b] | | | |
| Yes | 0 | 13 | 0.002 |
| No | 10 | 9 | |

**Comparison between cases with SSNS and SRNS
[a]The numbers include two sets of siblings with FSGS
[b]Includes dialysis and/or transplantation

synonymous variants have been previously reported and are predicted to be benign. A242V is a common polymorphism (frequency in all cases (including sib-pairs) (5/32) and controls (9/50), respectively). The previously reported exon 6 V260E pathogenic variant was present only in SRNS cases and therefore exon 6 was sequenced in 50 black controls to determine its frequency in unaffected individuals.

**NPHS2 associations with FSGS**. The *NPHS2* V260E variant is a nucleotide change from A to T that results in an amino acid change from valine to glutamic acid. The *NPHS2* variant was present in the homozygous state in 13/32 cases (this included the two sib-pairs) and in the heterozygous state in 4/32 FSGS cases and specifically associated with steroid resistance (Table 4) (Supplementary Fig. 1). None of the controls were homozygous for *NPHS2* V260E, however one black healthy control was heterozygous for the *NPHS2* V260E variant. The unrelated FSGS cases were grouped into SRNS (20/30) (67%) and SSNS (10/30) (33%). There was a significant difference in genotype frequencies between the unrelated steroid resistant cases (11/20 E/E and 4/20 V/E) and controls ($p = 2.07e-10$).

Table 5 shows the haplotypes associated with the *NPHS2* V260E variant in the 20 unrelated SRNS cases with the V260E variant. Of the 26 *NPHS2* 260E-associated alleles (including only one individual of each sib-pair), 24 were compatible with the TTGCCCAC haplotype, and two had alternate 260E-associated haplotypes (CGGTCCAC and TGGCCCAC). These results suggest a single origin for the V260E allele with some haplotype decay, likely arising from recombination events, giving rise to two alternate haplotypes.

**NPHS2 and APOL1 association with age of onset**. The V260E variant was not observed in any of the SSNS cases (Fig. 1A) and none had 2 *APOL1* risk alleles. *NPHS2* V260E was present in the homozygous state in 13/22 and heterozygous state in 4/22 of the SRNS cases, including both individuals from the two sib-pairs (Fig. 1B). Individuals with 2 *APOL1* risk alleles tended to have an earlier age of onset.

**Kidney survival for NPHS2 V260E variant**. A time-to-event analysis was performed for V260E by measuring the time from biopsy-proven diagnosis to onset of ESKD, and comparing individuals who had SSNS to those with SRNS (separating them into two groups—0 vs. 1 or 2 *NPHS2* V260E variants). Survival curves were combined for individuals with 1 and 2 *NPHS2* V260E variants, as there was no significant difference between the two ($p = 0.209$). The Kaplan–Meier plot depicts the kidney survival among 32 paediatric patients with FSGS. Out of the 32 FSGS cases, 13 progressed ESKD. A mean kidney survival of 69.2 and 26.8 months was observed when the V260E variant was absent and present in SRNS cases, respectively. There was a significant difference in kidney survival among the three groups ($p = 0.026$) (Fig. 2).

**Two families with multiple cases of SRNS**. The pedigrees of families A and B are shown in Fig. 3. Notably the age of a biopsy-confirmed FSGS differed by 7 years for the two siblings from family A who are homozygous for V260E. Proband II.3 was 3-years-old when a renal biopsy-confirmed FSGS and a renal transplant was performed at the age of 7 years. Proband II.2 was 10-years-old at the time of biopsy-confirmed FSGS. The mother and father each had one high risk *APOL1* allele. The mother passed on 1 high risk allele (G1) to two of her children, one of whom was unaffected (proband II.2). Proband II.3 inherited two low risk alleles, one from each parent. Both parents were heterozygous for the V260E mutation. In Family B, twins were affected with FSGS (SRNS). Proband II.2 was 6-years-old when FSGS was confirmed, and is currently on dialysis. Proband II.3 was 9-years-old when diagnosed with FSGS. DNA samples were unavailable for the unaffected children. The mother had 1 high risk *APOL1* allele, which she passed on to both affected children. Both parents were heterozygous for V260E, and each passed on the mutation to their affected children who were homozygous for V260E.

**Table 2 Association of *APOL1* renal risk variants with FSGS**

| Number risk alleles present | SSNS N = 10 | All FSGS w/o *NPHS2* V260E N = 15 | Controls N = 176 |
|---|---|---|---|
| 0 | 3 | 6 | 117 |
| 1 | 7 | 8 | 47 |
| 2 | 0 | 1 | 12 |
| 1 or 2 vs. 0 | 4.63, P = 0.03 (CI[a], 1.15–18.5) | 2.97, P = 0.048 (CI[a], 1.01–8.75) | Reference |

[a]CI = 95% confidence interval
w/o without

**Table 3 Variants in the *NPHS2* gene in unrelated black South African paediatric FSGS cases (N = 30)**

| Variant characteristics | | | | SS-FSGS N = 10 | | | SR-FSGS N = 20 | | | MAF according to the 1000 Genomes Project combined data | | |
|---|---|---|---|---|---|---|---|---|---|---|---|---|
| Exon | SNP ID | cDNA | Protein | Het | Hom | Allele Freq | Het | Hom | Allele Freq | Eur | Afr | Asian |
| 1 | rs78541594 | | | 1 | 0 | 0.05 | 2 | 0 | 0.05 | 0.003 | 0.230 | 0 |
| 1 | rs12406197 | | | 3 | 0 | 0.15 | 7 | 9 | 0.63 | 0.239 | 0.133 | 0.119 |
| 1 | rs1079292 | c.102A>G | p.Gly34Gly | 3 | 5 | 0.65 | 3 | 16 | 0.88 | 0.008 | 0.177 | 0.019 |
| 1 | rs111306764 | c.144C>T | p.Ser48Ser | 3 | 0 | 0.15 | 2 | 0 | 0.05 | | | |
| 2 | rs3738423 | c.288C>T | p.Ser96Ser | 1 | 0 | 0.05 | 1 | 0 | 0.03 | 0.087 | 0.094 | 0.095 |
| 4 | COSM900289 | | | 0 | 0 | 0 | 1 | 0 | 0.03 | – | – | – |
| 5 | rs61747727 | c.725C>T | p.Ala242Val[a] A242V | 1 | 0 | 0.05 | 4 | 0 | 0.10 | 0 | 0.073 | 0 |
| 6 | rs775006954 | c.779T>A | p.Val260Glu[b] V260E | 0 | 0 | 0 | 4 | 11 | 0.65 | – | – | – |
| 8 | rs1410592 | c.954C>T | p.Ala318Ala | 6 | 2 | 0.50 | 9 | 0 | 0.23 | 0.615 | 0.595 | 0.491 |
| 8 | rs3818587 | c.1038A>G | p.Leu346Leu | 1 | 0 | 0.05 | 1 | 0 | 0.03 | 0.087 | 0.095 | 0.094 |

[a]SIFT and PolyPhen predict Ala242Val is deleterious and probably damaging, respectively
[b]SIFT and PolyPhen predict Val260Glu is damaging and probably damaging, respectively

## Discussion

Among black South African children with biopsy proven FSGS, SRNS is more common than SSNS, presenting a challenge to clinicians when choosing appropriate therapy. We identified a *NPHS2* V260E variant that was present in 15 of the 20 unrelated SR-FSGS cases among black children, 11 in the homozygous and four in the heterozygous state. An autosomal recessive mode of inheritance is supported by the two familial cases. In the familial cases the age of disease onset was different, despite the homozygosity for the V260E mutation, suggesting variable expressivity and the presence of potential environmental and genetic modifiers.

Interestingly there were four V260E heterozygotes, of whom two were also heterozygous for the A242V variant, which although thought to be a functionally benign polymorphic variant, may exert an effect in the compound heterozygote state. The *NPHS2* V260E variant has a population frequency of 0.002% in individuals of African descent in the ExAC database and has not been detected in

Europeans, Asians, or African Americans. This variant was previously observed in the homozygous state in several consanguineous families with SRNS and in unrelated children from KwaZulu-Natal with sporadic FSGS[20,27–29]. Children homozygous for the variant had SR-FSGS and did not respond to steroid therapy, consistent with findings for the disease severity of V260E[21,27]. Children with two *NPHS2* V260E variants and two *APOL1* risk alleles appear to develop SRNS at a younger age (Fig. 1B), but more data are required to confirm this finding.

The *NPHS2* gene encodes the protein podocin that interacts with nephrin (encoded by *NPHS1*) and CD2-associated protein (encoded by *CD2AP*) to maintain the structural integrity of the slit diaphragm that acts as a filtration barrier in the kidney[21]. Previous studies identified *NPHS2* variants as the most common cause of autosomal recessive SRNS in childhood[8,9]. Although most of the *NPHS2* mutations cause childhood SRNS, some cause SRNS in adulthood, for example *NPHS2* V290M was detected in late-onset SRNS in central and eastern Europeans[30]. Our study found a total of 10 variants in the *NPHS2* gene, all of which were previously described and of which eight were predicted to be benign (Table 3). Variant A242V, found in exon 5, was predicted to be deleterious and probably damaging by SIFT and PolyPhen, respectively. A previous study showed that A242V was one of the most common variants found in a FSGS patient cohort and it was not associated with proteinuria, and did not appear to cause FSGS[31]. The ExAC browser reports 29 homozygotes for the A242V variant, and it has a minor allele frequency (MAF) of 0.07 in African descent populations and is not present in European or Asian populations of the 1000 Genomes Project. Since the MAF of A242V in Africans tends to be greater than 5%, it is unlikely that this variant has a severe pathogenic effect on its own.

**Table 4 *NPHS2* V260E in unrelated black South African paediatric FSGS cases (*N* = 30)**

| *NPHS2* V260E | SSNS *N* = 10 | SRNS *N* = 20 | Controls *N* = 50 |
|---|---|---|---|
| *Genotypes* | | | |
| V/V | 10 (100%) | 5 (25%) | 49 (98%) |
| V/E | 0 | 4 (20%) | 1 (2%) |
| E/E | 0 | 11 (55%) | 0 |
| *Allele frequency* | | | |
| V | 1 | 0.35 | 0.99 |
| E | 0 | 0.65 | 0.01 |
| *SRNS VS. controls* | | | |
| | | **P** = 2.07e-10 | |

**Table 5 Different haplotypes identified in unrelated SRNS individuals with the V260E variant**

| | rs115256710 T/C | rs12406197 G/T | rs1079292 A/G | rs111306764 C/T | COSM900289 C/A | rs61747727 C/T | rs775006954 T/A[a] | rs141 0592 C/T | E chromosomes *N* = 26 |
|---|---|---|---|---|---|---|---|---|---|
| Haplotype 1 | T | T | G | C | C | C | A | C | 24/26 |
| Haplotype 2 | C | G | G | T | C | C | A | C | 1/26 |
| Haplotype 3 | T | G | G | C | C | C | A | C | 1/26 |

Affected siblings were also E/E and are not included in the table
[a]The *NPHS2* V260E variant that results in an amino acid change from valine to glutamic acid

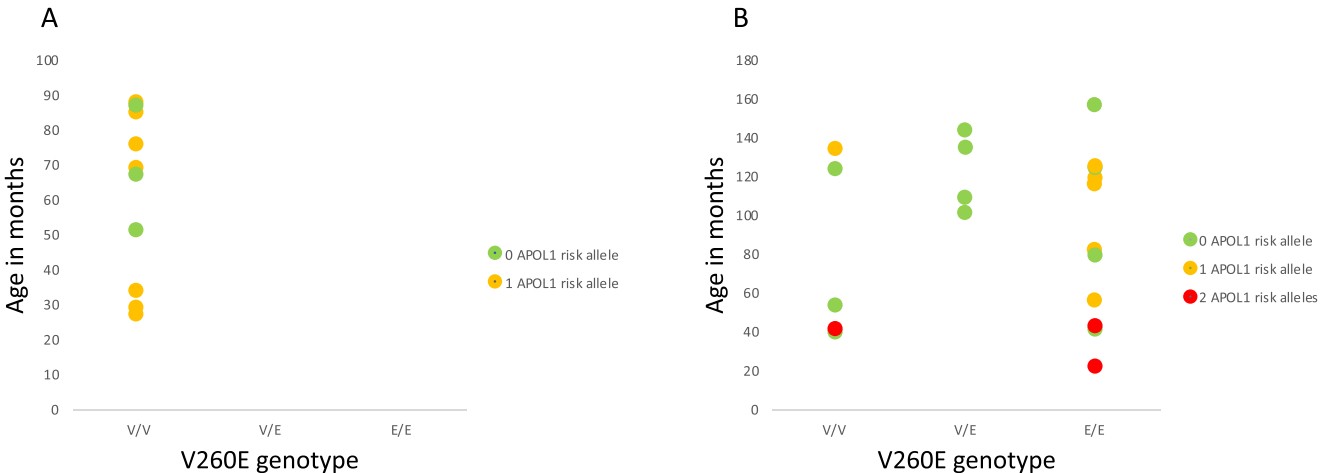

**Fig. 1 A** Comparison of SSNS cases (*n* = 10) and **B** SRNS cases (*n* = 22) (including sib-pairs) showing *NPHS2* V260E genotypes, age of onset (vertical axis) and colour coding according to number of *APOL1* risk alleles

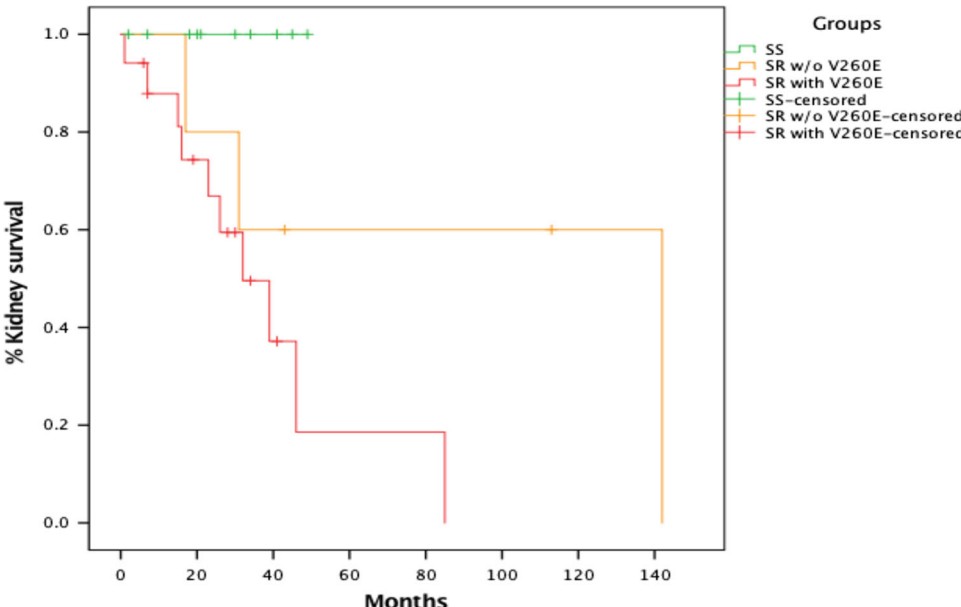

**Fig. 2** Kaplan–Meier plot showing kidney survival from biopsy-proven diagnosis to renal failure (determined by either the start of dialysis, at the time of kidney transplant or individuals with eGFR < 15 ml/min/1.73 m$^2$) for individuals with SSNS (0 *NPHS2* V260E variant), SRNS (0 *NPHS2* V260E variant) and SRNS (1 or 2 *NPHS2* V260E variants). Vertical lines indicate duration of observation from the time of diagnosis in individuals who have not yet reached renal failure

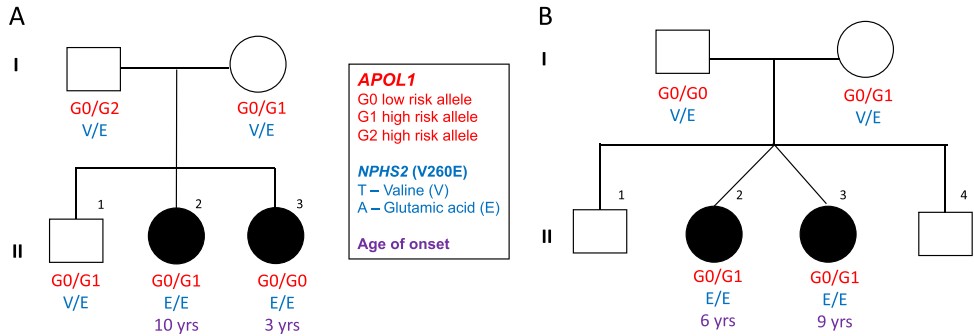

**Fig. 3** Pedigrees showing segregation of *APOL1* alleles (G0, G1 and G2) and the *NPHS2* V260E variant in the two familial cases. Age of onset of renal disease is shown. In family B, blood samples were not available for the unaffected siblings

The Kaplan–Meier plot shows the V260E variant appears to be associated with a decline in kidney function, with a modest follow-up of 60 months (Fig. 2). The censored markings (short vertical lines) on the graph depict the cases in which ESKD had not yet occurred during the time we observed the individual, and therefore the total number of months during which the individual did not develop ESKD is shown. The plot suggests that individuals who did not have the *NPHS2* V260E variant progressed more slowly to ESKD; the groups differed significantly according to the log rank test ($p = 0.026$).

The high prevalence of FSGS among individuals of African ancestry in America has been partly attributed to the presence of *APOL1* risk variants[16,17,32], however, the association with FSGS was not observed among a small group of black adult South Africans with FSGS[12]. In our study, the combined G1 and G2 risk allele frequency among paediatric FSGS cases was 32%. This was higher than the 18% observed in adult black South Africans with HIV negative CKD by Kasembeli et al. (2015) and lower or similar to the reported frequencies in West Africa (Yoruba from Nigeria ~45%; Igbo from Niger ~30%) and African Americans (~34%)[18,33–35]. Smith and Malik (2009) and Limou et al. (2014) hypothesised that in the last 10,000 years, *APOL1* variants arose

in sub-Saharan Africans, most likely in West Africans, where they were subjected to strong recent positive selection likely due to the presence of trypanosomiasis[36,37]. There is evidence of a recent selective sweep, seen as extended haplotype homozygosity in the region harbouring the *APOL1* variants, observed only in Yoruba individuals from Nigeria[16,37]. The lower frequency in our study suggests a possible relaxation of selection pressures once the population migrated out of a trypanosomiasis endemic region, or gene flow from West Africans into the gene pool that followed the Bantu expansion to east and south Africa[18]. Due to the low frequency of the G1 and G2 risk alleles in South African populations, our study was underpowered to examine the role of *APOL1* risk alleles in the progression of kidney disease.

The *NPHS2* V260E variant in the two familial cases followed an autosomal recessive inheritance pattern and the *APOL1* risk variants did not appear to influence age of onset in these families. Further investigation would be necessary to understand why the affected siblings were diagnosed with FSGS at different ages and progressed at different rates, despite the fact that the monozygotic twins shared similar environmental exposures and genetic background. Since the V260E mutation is common among South African children with SRNS, it could be used as a diagnostic

marker in children with familial or sporadic FSGS to avoid immunosuppressant therapy that is likely to result in adverse effects, with no benefit to the patient[28,38].

The limitations of the study include its modest size and the fact that the age of onset is reflected as the age at biopsy confirmed diagnosis and may, in fact, have been considerably earlier. Since the study was under-powered, we could not definitively confirm the association of two *APOL1* risk alleles with a younger age of disease onset.

Our results strengthen and expand the association of the *NPHS2* V260E variant with SRNS in black South African children from different ethnic origins, including Tsonga, Tswana, Swati, Sotho, Pedi and Zulu. The major 260E-associated haplotype suggests that it has a single origin. Screening all black African children with FSGS for *NPHS2* V260E has the potential to be highly predictive of a diagnosis of SRNS. Identifying patients who would not respond well to immunosuppressants would reduce morbidity and prevent the need for invasive renal biopsies. It could also be used for genetic counselling to provide accurate recurrence rates for affected families.

## Methods

**Participants**. Study participants included children between the ages of 2 and 17 years with FSGS who were recruited from the Division of Paediatric Nephrology of Charlotte Maxeke Johannesburg Academic Hospital (CMJAH) and Morningside Mediclinic (MM), both in Johannesburg. Only children with biopsy-proven FSGS, younger than 18 years at diagnosis, self-reported black African ethnicity, adequate data in their clinical files and accessible for participation were included. Children with HIV-associated glomerular disease on biopsy were excluded. The cases were sporadic, with the exception of two pairs of affected siblings whose nuclear families (parents and children) were included in the study.

The controls from the Sydney Brenner Institute for Molecular Bioscience (SBIMB) biobank included 226 DNA samples of similar geographic and ethnic origins. Of these, 176 were included in the *APOL1* risk variant association analysis and 50 for comparison with *NPHS2* variants. All participants assented to participate in the study and their parents or caregivers provided informed consent according to the documents approved by the Human Research Ethics Committee (Medical) (Approval code: M170657).

**Data capture**. The following information was extracted from patient files and entered into a REDCap database[24]: demographic information, patient follow-up details, medical history, baseline and annual follow-up information, height, weight and serum creatinine values with corresponding dates. Longitudinal data were used to assess rate of disease progression. This was achieved by using serum creatinine and height to estimate glomerular function rate using the Bedside Schwartz equation[11]. Glomerular filtration rate (GFR) was calculated as follows; GFR (ml/min/1.73 m$^2$) = 0.413 × (height in cm/serum creatinine in μmol/l). All identifying data were removed and codes were provided prior to exporting the data for analysis to ensure anonymity and maintain confidentiality.

**DNA extraction and sequencing**. Genomic DNA was extracted from peripheral blood samples using the salting-out procedure[25]. The concentration and quality of the DNA samples were examined using a NanoDrop spectrophotometer. The genotypes of the three *APOL1* variants were obtained using Sanger sequencing and the following primers: forward – 5′TCAGCTGAAAGCGGTGAACA3′ and reverse − 5′GGCATATCTCTCCTGGTGGC3′. The *APOL1* variants comprise two non-synonymous variants collectively referred to as the G1 allele (actually a haplotype) (rs73885319 and rs60910145) and a 6 bp in-frame deletion (rs71785313) referred to as the G2 allele. *APOL1* therefore has a low risk allele (G0) and 2 high risk alleles (G1 and G2). Exons 1 to 8 of the *NPHS2* gene were sequenced on the ABI 3500 XL as per manufacturer protocol using primers from Boute et al. (2000) for exons 2–7[26]. A new primer was designed for *NPHS2* exon 1 (forward – 5′CCAGAGCTT GCGATGAGCTTCTGTATC3′, reverse 5′CCGTTCCTGGGAACCTGAGCATC CAGC3′).

**Statistics and reproducibility**. The association between *APOL1* risk alleles and FSGS was tested by using Fisher's exact test to compare frequencies of *APOL1* risk alleles in all cases and controls. Odds ratios (ORs) with 95% confidence intervals (CIs) where calculated for the association between *APOL1* risk alleles (0 versus 1 or 2 risk alleles) and FSGS.

Fisher's exact test was also used to compare the frequency of V260E in SRNS versus controls. A non-parametric Kaplan–Meier plot was used to assess progression from renal biopsy to kidney failure (dialysis and/or transplant and ≤15 ml/min/1.73 m$^2$).

**Reporting summary**. Further information on research design is available in the Nature Research Reporting Summary linked to this article.

## Data availablility

All data are presented in the paper and requests for clarification can be made to the corresponding author.

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

## Acknowledgements

We are grateful to the patients, their families and our control participants for taking part in the study and to the nurses and doctors at the clinics who generously assisted us. We thank Cassandra Soo and her team at the SBIMB biobank for preparing the DNA samples and also thank Professor Tikly and Drs Govind and van Hougenhoek-Tulleken for sharing their *APOL1* data for the controls. MR is a South African Research Chair in Genomics and Bioinformatics of African populations hosted by the University of the Witwatersrand, funded by the Department of Science and Technology and administered by National Research Foundation of South Africa (NRF). This study was funded by the chair.

## Author contributions

G.A.M., F.J., R.M. and W.A.C. designed the study. M.H. and G.A.M. collected data from clinics in collaboration with G.E., L.C. and M.G. Data analysis and writing of the paper was done by Govender M. R.M. and F.J. contributed to guiding analysis and extensively editing the paper. All other authors approved the paper.

## Competing interests

The authors declare no competing interests.

## Additional information

**Supplementary information** is avalibale for this paper at https://doi.org/10.1038/s42003-019-0658-1.

