## [Peer Review File · Communications Biology]

Reviewers' comments:

Reviewer #1 (Remarks to the Author):

Govender et al. studied 30 families with childhood FSGS, 20 of whom presented with steroid-resistant (SRNS) and 10 with steroid-sensitive nephrotic syndrome (SSNS). They sequenced NPHS2, the major gene of monogenic SRNS and genotyped the APOL1 risk alleles, which are known to contribute to polygenic forms of the disease.

They found the well-known NPHS2 V260E mutation in the homozygous state in 11/20 SRNS families and in the heterozygous state in 4 of the remaining 9 families. Children with homozygous V260E progressed to ESRD by the age of 7 years.

Among the 15 children without V260E mutation (10 SSNS and 5 SRNS) they found the APOL1 risk alleles to be more common than among the 176 controls (OR: 2.97, CI: 1.01-8.75).

The V260E mutation is well known and frequent founder mutation in South Africa. As cited by the authors (Asharam et al, 2018) it was recently reported to be very frequent in SRNS, in accordance with the authors' data. It is well known that NPHS2 mutations cause an autosomal recessive form of SRNS, there is no need to conclude it in the first paragraph of the discussion. The transmission of the V290M mutation is incorrectly cited, it was found in late-onset recessive SRNS (Kerti et al, 2013). As the NPHS2-related SRNS is a completely penetrant monogenic disorder, it seems useless to calculate the OR of patients with homozygous V260E to develop SRNS (Table 4 and Page 7, last paragraph).

The second message of the work is the higher number of APOL1 risk alleles in the 15 patients with NS but without V260E, suggesting that the APOL1 risk alleles contribute to the development of childhood NS. This conclusion is not convincing for several reasons:

- the number of patients is very low, as well as the statistical significance
- the APOL1 risk alleles significantly increase the risk of nephropathy when present on both alleles (Genovese et al, Science, 2010, Kipp et al, JASN, 2011). In this study, patients with one (n=8) or two risk alleles (n=1) were grouped and compared to controls together. This should be discussed.
- the work of Asharam et al, 2018, also co-authored by one of the authors of the present paper, concluded that there is no association between NS and the APOL1 variants
- the paper states that the patients were of different ethnic origin (page 7, 1st paragraph). It is also discussed that the APOL1 variants have a highly variable allele frequency in the different populations of Africa. Does the ethnic origin of the control population correspond to that of the patients? The role of different ethnicities is also suggested by the APOL1 genotype of patients with homozygous V260E (Figure 1). These 11 patients have an even higher proportion of APOL1 risk alleles (9/22 alleles) than the 15 patients without V260E (10/30 alleles), though their SRNS is explained completely by the NPHS2 V260E mutation.

The paper should emphasize the high frequency of the V260E mutation in the South African SRNS patients and conclude the role of the APOL1 risk alleles with caution.

Minor comment:

Page 2, 2nd paragraph: „Genes commonly mutated in childhood SRNS are nephrin...“ Nephrin, podocin are not genes.

Reviewer #2 (Remarks to the Author):

Authors have investigated Genetics of Nephrotic syndrome, specifically Focal Segmental Glomerulosclerosis in group of African Children, with targeted genes variants.

The phenotyping validation of the diagnosis by renal biopsy is a strength of the study, as well as the prospective design. The study confirms the important of APOL1 risk variants in this kidney disease in African children and revealed the implication of variant in NPHS2 in FSGS with rapid progression to end stage kidney disease in this group of South African Black children.

The study will add to the literature, with the most needed genetic data from the African populations, however could be improved with a few clarifications:

Major comment

The authors stressed on the stronger association of variants in NPHS2 with progression to ESKD; I

will suggest toning down this statement, because of the modest period of prospective follow-up in some cases (60 months). In addition, the low frequency of APOL1 risk alleles in the South African Population suggests that the sample size of the patients' population investigated is not sufficiently powered to properly evaluate the implication of APOL1 risk alleles in the progression of ESKD.

Minor Comments

- Authors should provide some background on the toll Kidney disease in South African populations to put the study in Context; as well the availability of genetic services, since the authors suggested the use of the results for genetic counselling purpose.
- Were HWE test performed for variants?
- Owing to the modest sample size they may be some value to provide a few data on the cases that were loss in follow up: genotypes and association with KD with cross-sectional approach
- The length of the discussion can be shortened

Reviewer #3 (Remarks to the Author):

In this paper by Govender et al titled "Common African-specific NPHS2 V260E mutation in steroid resistant nephrotic syndrome in black South African children with biopsy proven FSGS", the authors reported that the previously reported V260E variant (Asharam K, et al. NPHS2 V260E Is a Frequent Cause of Steroid-Resistant Nephrotic Syndrome in Black South African Children. *Kidney Int Rep.* 2018 Jul 29;3(6):1354-1362) in NPHS2 is a leading cause of monogenic FSGS. As expected being homozygous for V260E was associated with 100-fold increase in risk of SRNS and progression of disease. This study confirmed the findings from the previous study.

Reviewer #1

It is well known that NPHS2 mutations cause an autosomal recessive form of SRNS, there is no need to conclude it in the first paragraph of the discussion.

Response: Thank you for the suggestion. The sentence was edited accordingly in the first paragraph of the discussion on page 8 .

The transmission of the V290M mutation is incorrectly cited, it was found in late-onset recessive SRNS (Kerti et al, 2013).

Response: Thank you for pointing this out. The sentence was edited to include the AR transmission of the V290M mutation in SRNS cases in the second paragraph on page 9 .

As the NPHS2-related SRNS is a completely penetrant monogenic disorder, it seems useless to calculate the OR of patients with homozygous V260E to develop SRNS (Table 4 and Page 7, last paragraph).

Response: We agree with this statement. The OR calculations were removed from the abstract, the second paragraph on page 7 and Table 4.

The second message of the work is the higher number of APOL1 risk alleles in the 15 patients with NS but without V260E, suggesting that the APOL1 risk alleles contribute to the development of childhood NS. This conclusion is not convincing for several reasons:

- the number of patients is very low, as well as the statistical significance

Response: We agree with this statement and have moderated this point to reflect the uncertainty in the first paragraph on page 10.

- the work of Asharam et al, 2018, also co-authored by one of the authors of the present paper, concluded that there is no association between NS and the APOL1 variants

Response: Our study differed from Asharam et al, 2018 in that they only had 1 black individual with SS-FSGS and therefore results are not comparable.

- the paper states that the patients were of different ethnic origin (page 7, 1st paragraph). It is also discussed that the APOL1 variants have a highly variable allele frequency in the different populations of Africa. Does the ethnic origin of the control population correspond to that of the patients? The role of different ethnicities is also suggested by the APOL1 genotype of patients with homozygous V260E (Figure 1). These 11 patients have an even higher proportion of APOL1 risk alleles (9/22 alleles) than the 15 patients without V260E (10/30 alleles), though their SRNS is explained completely by the NPHS2 V260E mutation.

Response: Yes. The sentence was edited in the second paragraph of the methods on page 4 .

“Page 2, 2nd paragraph: „Genes commonly mutated in childhood SRNS are nephrin...”
Nephrin, podocin are not genes”.

Response: The sentence was edited for clarity in the first paragraph on page 3. However, according to HUGO Gene Nomenclature Committee (<https://www.genenames.org/>) the gene names for *NPHS1* and *NPHS2* are nephrin and podocin, respectively.

Reviewer #2

The authors stressed on the stronger association of variants in NPHS2 with progression to

ESKD; I will suggest toning down this statement, because of the modest period of prospective follow-up in some cases (60 months).

Response: Thank you for the suggestion. The statement was edited to add less emphasis on the association between the V260E variant and progression to ESKD in the third paragraph on page 9.

In addition, the low frequency of APOL1 risk alleles in the South African Population suggests that the sample size of the patients' population investigated is not sufficiently powered to properly evaluate the implication of APOL1 risk alleles in the progression of ESKD.

Response: Thank you for the comment. A sentence was added in the first paragraph on page 10 stating the study was underpowered to examine the role of *APOL1* risk alleles in the progression of kidney disease.

Authors should provide some background on the toll Kidney disease in South African populations to put the study in Context; as well the availability of genetic services, since the authors suggested the use of the results for genetic counselling purpose.

Response: Thank you for the comment. Background information was added in the introduction in the first paragraph on page 4.

Were HWE test performed for variants?

Response: Yes, HWE p values for G1 and G2 in the cases were 0.52 and 0.54.

Owing to the modest sample size they may be some value to provide a few data on the cases that were loss in follow up: genotypes and association with KD with cross-sectional approach.

Response: We tried to reach individuals using last known phone numbers but to no avail. Hence, we could not obtain blood samples for genotyping.

"The length of the discussion can be shortened"

Response: We have attempted some shortening.